# Effects of dietary intake and nutritional status on cerebral oxygenation in patients with chronic kidney disease not undergoing dialysis: A cross-sectional study

Susumu Ookawara[1,2]☯*, Yoshio Kaku[1]☯, Kiyonori Ito[1], Kanako Kizukuri[2], Aiko Namikawa[2], Shinobu Nakahara[2], Yuko Horiuchi[2], Nagisa Inose[2], Mayako Miyahara[2], Michiko Shiina[2], Saori Minato[1], Mitsutoshi Shindo[1], Haruhisa Miyazawa[1], Keiji Hirai[1], Taro Hoshino[1], Miho Murakoshi[2], Kaoru Tabei[3], Yoshiyuki Morishita[1]

1 Division of Nephrology, First Department of Integrated Medicine, Saitama Medical Center, Jichi Medical University, Saitama, Japan, 2 Department of Nutrition, Saitama Medical Center, Jichi Medical University, Saitama, Japan, 3 Department of Internal Medicine, Minami-uonuma City Hospital, Niigata, Japan

☯ These authors contributed equally to this work.
* su-ooka@hb.tp1.jp

**Data Availability Statement:** All relevant data are within the paper.

## Abstract

### Background

Dietary management is highly important for the maintenance of renal function in patients with chronic kidney disease (CKD). Cerebral oxygen saturation ($rSO_2$) was reportedly associated with the estimated glomerular filtration rate (eGFR) and cognitive function. However, data concerning the association between cerebral $rSO_2$ and dietary intake of CKD patients is limited.

### Methods

This was a single-center observational study. We recruited 67 CKD patients not undergoing dialysis. Cerebral $rSO_2$ was monitored using the INVOS 5100c oxygen saturation monitor. Energy intake was evaluated by dietitians based on 3-day meal records. Daily protein and salt intakes were calculated from 24-h urine collection.

### Results

Multivariable regression analysis showed that cerebral $rSO_2$ was independently associated with energy intake (standardized coefficient: 0.370) and serum albumin concentration (standardized coefficient: 0.236) in Model 1 using parameters with $p < 0.10$ in simple linear regression analysis (body mass index, Hb level, serum albumin concentration, salt and energy intake) and confounding factors (eGFR, serum sodium concentration, protein intake), and the energy/salt index (standardized coefficient: 0.343) and Hb level (standardized coefficient: 0.284) in Model 2 using energy/protein index as indicated by energy intake/protein intake and energy/salt index by energy intake/salt intake in place of salt, protein and energy intake.

**Funding:** This work was supported by a grant from the Japanese Association of Dialysis Physicians (http://www.touseki-ikai.or.jp/) (JADP Grant 2017-9) and a grant from The Kidney Foundation, Japan (JKFB 17-4) (http://www.jinzouzaidan.or.jp/) to SO. The funders had no role in study design, data collection and analysis, decision to publish, or preparation of the manuscript.

**Competing interests:** The authors have declared that no competing interests exist.

## Conclusions

Cerebral rSO$_2$ is affected by energy intake, energy/salt index, serum albumin concentration and Hb level. Sufficient energy intake and adequate salt restriction is important to prevent deterioration of cerebral oxygenation, which might contribute to the maintenance of cognitive function in addition to the prevention of renal dysfunction in CKD patients.

## Introduction

Diet therapy, including the energy intake management and protein and salt restriction, is a key aspect of chronic kidney disease (CKD) therapy and makes an important contribution to the maintenance of renal function. Several important guidelines have been proposed regarding the dietary intake of CKD patients. In the clinical setting of CKD management in Japan, energy intake is recommended to be within 25–35 kcal/kg ideal body weight (BW) [1–3] and protein intake is recommended to be 0.6–1.0 g/kg ideal BW [1,4–6]. These recommendations differ according to the stage of CKD, and a salt intake of 3–6 g/day is suggested to be ideal [1,7,8]. Low energy intake has been reported to be associated with deterioration of renal function [9,10], and increased salt intake could increase the risk of progression of renal dysfunction in CKD patients [11,12].

Recently, near-infrared spectroscopy (NIRS) has been used as a tool to measure the regional saturation of oxygen (rSO$_2$), a marker of tissue oxygenation, in order to clarify the influence of CKD progression on cerebral oxygenation in CKD patients receiving hemodialysis (HD) [13–17]. The results of these measurements reflect the status of cognitive impairment because of the relationship of rSO$_2$ with the Mini-Mental State Examination scores [16] and the Montreal Cognitive Assessment test [17]. Furthermore, cerebral rSO$_2$ has been shown to decrease with decreasing estimated glomerular filtration rate (eGFR) [17]. Therefore, cerebral rSO$_2$ may be influenced by the nutritional status of CKD patients, because of the impact of dietary intake on renal function. To date, few reports have investigated the relationship between cerebral oxygenation using NIRS and dietary intake in CKD patients who are not receiving dialysis therapy, and data regarding the association between cerebral rSO$_2$ and nutritional status of such patients is limited. This study aimed to investigate the influence of dietary intake and nutritional status on the cerebral oxygenation of CKD patients not receiving dialysis therapy.

## Materials and methods

### Patients

In this single-center observational study, CKD patients who met the following criteria were enrolled: (1) all-stage CKD patients not yet requiring dialysis who were followed up in the Division of Nephrology of our hospital, (2) patients who were older than 20 years, (3) patients who received dietary education and nutritional assessment for CKD management, and (4) patients who underwent 24-hour urine collection for the evaluation of salt and protein intake. Exclusion criteria were the following comorbidities: congestive heart failure, chronic obstructive pulmonary disease, apparent neurological disorder, or chronic hypotension (defined as systolic blood pressure <100 mmHg). Fig 1 shows the flow chart of patient enrollment and analysis.

Sixty-seven patients were included in this study (47 men, 20 women; mean age, 65.6 ± 15.6 years). As shown in Table 1, the numbers of patients at each CKD stage were as follows: G1, 1;

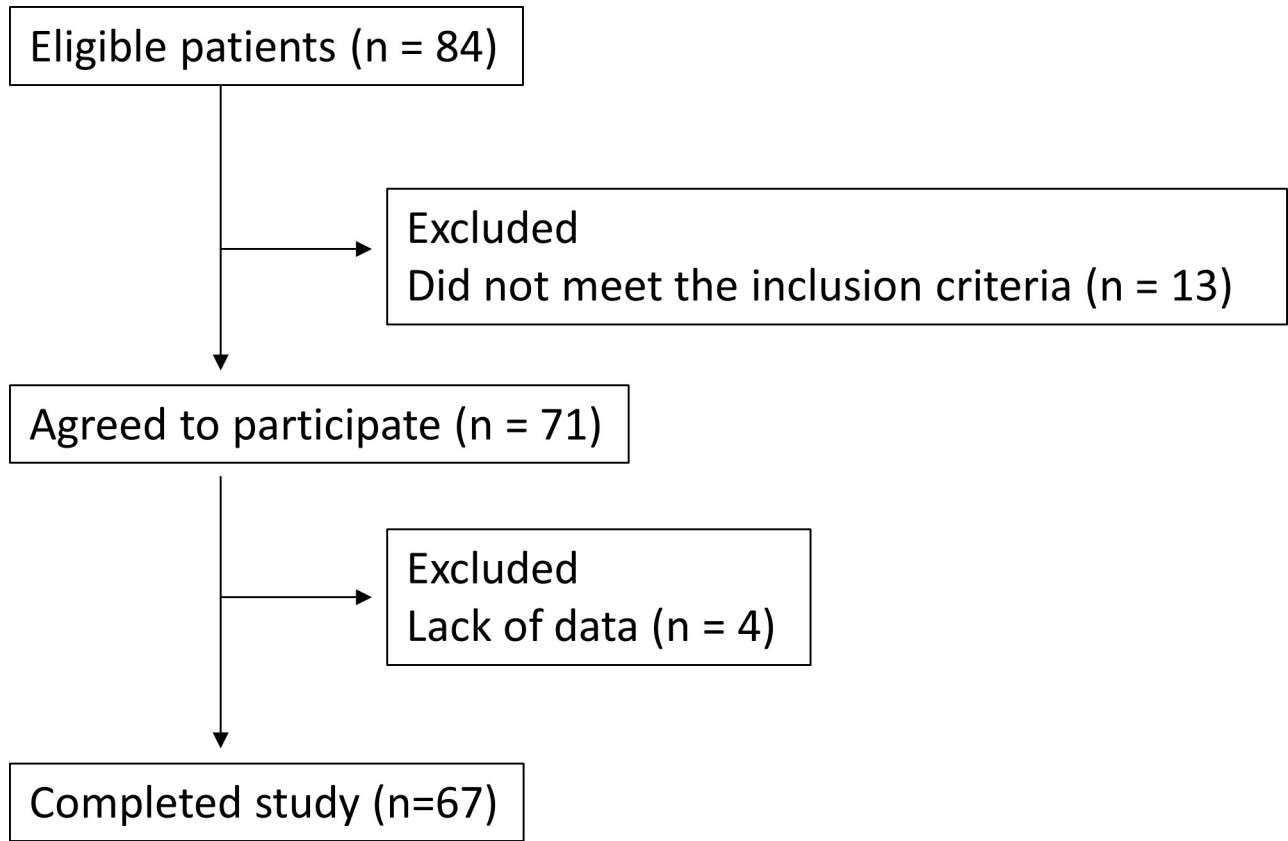

**Fig 1. Patient flow chart.**

G2, 1; G3a, 6; G3b, 12; G4, 28; and G5, 19. Causes of chronic renal failure included type 2 diabetes mellitus (32 patients), nephrosclerosis (19 patients), chronic glomerulonephritis (eight patients), and other causes (eight patients). All patients provided written informed consent to participate in this study. This study and its protocols were approved by the Institutional Review Board of Saitama Medical Center, Jichi Medical University, Japan (DAI-RIN 15–104) and conform to the provisions of the Declaration of Helsinki (as revised in Tokyo in 2004).

### Evaluation of patient's renal function

For the classification of CKD stages, renal function was evaluated using eGFR based on the serum creatinine concentration (S-Cr), and eGFR was calculated using Eq 1 [18]:

$$\text{eGFR (mL/min/1.73 m}^2) = 194 \times \text{S--Cr}^{-1.094} \times \text{age}^{-0.287} \text{ (for men)}$$
$$\text{eGFR (mL/min/1.73 m}^2) = 194 \times \text{S--Cr}^{-1.094} \times \text{age}^{-0.287} \times 0.793 \text{ (for women).}$$

(1)

### Method of nutritional assessment

Patients included in this study were asked to record the total quantity of food and beverages consumed either by weight or in household measures and to record the methods of food preparation. Energy intake was evaluated by dietitians based on each patient's 3-day meal record using the fifth edition of the Japanese Standard Tables of Food Composition published by the Science and Technology Agency of Japan [19]. Furthermore, 24-h urine collection was performed to enable evaluation of urinary protein excretion (g/day), urinary urea nitrogen

**Table 1. Patient characteristics.**

| Characteristics | Total patients n = 67 |
|---|---|
| Male/female | 47/20 (70/30) |
| Cerebral rSO$_2$ (%) | 55.9 ± 6.6 |
| CKD stages G1/2/3a/3b/4/5 | 1 (1)/1 (1)/6 (9)/12 (18)/28 (42)/19 (28) |
| Disease | |
| Diabetes mellitus | 32 (48) |
| Nephrosclerosis | 19 (28) |
| Chronic glomerulonephritis | 8 (12) |
| Others | 8 (12) |
| Antihypertensive medication | |
| Renin-angiotensin system blocker | 41 (61.2) |
| Calcium channel blocker | 41 (61.2) |
| Beta blocker | 22 (32.8) |
| Diuretics (loop and/or thiazide) | 23 (34.3) |
| Antidiabetic medication | |
| Insulin agent | 9 (13.4) |
| Dipeptidyl peptidase-4 inhibitor | 17 (25.4) |
| Insulin secretagogue | 4 (6.0) |
| α-glucosidase inhibitor | 3 (4.5) |
| Thiazolidinedione | 3 (4.5) |
| Sodium-glucose cotransporter-2 inhibitor | 3 (4.5) |
| Others | |
| Vitamin D analog | 10 (14.9) |
| Phosphate binder | 6 (9.0) |
| Statin | 21 (31.3) |
| Antiplatelet agents | 19 (28.4) |
| Erythropoiesis-stimulating agent | 19 (28.4) |

Categorical data are presented as number (%), continuous data are presented as mean ± standard deviation.

Abbreviations: CKD, chronic kidney disease; rSO$_2$, regional oxygen saturation.

(UUN) excretion, and urinary Na$^+$ excretion. The urine collection method was as follows: collection was started in the morning after the first morning urine was discarded. Thereafter, the entire volume of urine was collected in a disposable 3L container. To avoid the possibility of inadequate urine collection, we trained all patients to properly collect their urine samples and emphasized that collection must be initiated at a specific time and completed at the same time the next day. Daily protein and salt intakes were calculated based on the UUN and urinary Na$^+$ excretion values obtained from the 24-h urine collection.

Protein intake was calculated using Maroni's equation [20], as described in Eq 2:

$$\text{Protein intake (g/kg ideal BW/day)} = (\text{BW (kg)} \times 0.031 + \text{UUN (g/day)}) \times 6.25 \div \text{ideal BW (kg)}. \tag{2}$$

Salt intake was calculated using Eq 3:

$$\text{Salt intake (g/day)} = \text{urinary Na}^+ \text{ excretion (mEq/day)} \div 17 \tag{3}$$

Furthermore, dietary education was provided by a dietician according to the protocols for nutritional management for CKD therapy in Japan; specifically, sufficient energy intake (25–35 kcal/kg ideal BW/day), protein restriction (0.6–1.0 g/kg ideal BW/day), and salt restriction (3–6 g/day) [1]. To evaluate the influence of energy intake, protein restriction, and salt restriction on cerebral oxygenation, we calculated the nutritional markers described in Eqs 4 and 5:

$$\text{Energy/protein index (kcal/kg ideal BW/g protein)} = \text{Energy intake} \\ \text{(kcal/kg ideal BW/day)} \div \text{Protein intake (g/day)} \tag{4}$$

$$\text{Energy/salt index (kcal/kg ideal BW/g salt)} = \text{Energy intake} \\ \text{(kcal/kg ideal BW/day)} \div \text{salt intake (g/day)} \tag{5}$$

## Cerebral oxygenation monitoring and clinical laboratory measurements

Cerebral $rSO_2$ was monitored using an INVOS 5100c saturation monitor (Covidien Japan, Tokyo, Japan), which utilizes NIRS technology. This instrument uses a light-emitting diode, which transmits near-infrared light at two wavelengths (735 and 810 nm), and two silicon photodiodes, which act as light detectors to measure oxygenated and deoxygenated hemoglobin (Hb). The ratio of the oxygenated to total Hb (i.e., oxygenated Hb + deoxygenated Hb) signal strength was read as a single numerical value that represents $rSO_2$ [21,22], and all data were immediately and automatically stored in sequence. The inter-observer variance for this instrument; namely, the reproducibility of the $rSO_2$ measurement, has been reported to be acceptable [23–25]. Therefore, $rSO_2$ is considered a reliable indicator for the estimation of actual cerebral oxygenation. Furthermore, the light paths leading from the emitter to the different detectors share a common part; the 30-mm detector assesses superficial tissues, while the 40-mm detector is used to assess deep tissues. By analyzing the differential signals recorded by the two detectors, the data for cerebral $rSO_2$ can be supposed to be obtained from deep tissue, 20–30 mm from the body's surface [26,27]. Before measurement, patients were asked to sit in the chair for at least 5 min, and an $rSO_2$ measurement sensor was attached to the patient's forehead. Thereafter, $rSO_2$ was measured at 6-s intervals for 5 min, and the mean value calculated. Blood and urinary samples were also obtained from each patient under ambient conditions. This measurement was performed approximately from 2 h to 4 h after each meal for each patient.

Clinical parameters including Hb, serum creatinine, sodium, potassium, chloride, total protein, serum albumin, urinary protein, urinary urea nitrogen, and urinary sodium concentration were measured in our hospital laboratory.

## Statistics

Data are expressed as mean ± standard deviation or median (interquartile range) as appropriate. Urinary protein excretion did not show normal distribution, and this variable was transformed using the natural log (ln). Correlations between cerebral $rSO_2$ and each clinical parameter, including nutritional parameters, were evaluated using Pearson's correlation coefficient and linear regression analysis. Variables with a p value below 0.10 in simple linear regression analysis and plausible confounding factors were included in multivariable linear regression analysis to identify factors affecting cerebral $rSO_2$ in CKD patients. Statistical significance was accepted at $p < 0.05$. All analyses were performed using SPSS Statistics for Windows, version 19.0 (IBM Corp., NY, USA).

## Results

The mean cerebral $rSO_2$ values of the CKD patients in this study were 55.9 ± 6.6%, and these were significantly positively correlated with Hb level, serum albumin concentration, energy intake, and energy/salt index. Cerebral $rSO_2$ was negatively correlated with body mass index (Table 2). Cerebral $rSO_2$ was negatively correlated with salt intake (r = -0.228, p = 0.064) and positively correlated with energy/protein index (r = 0.203, p = 0.099), although these correlations were not significant. Fig 2 illustrates the significant correlation between cerebral $rSO_2$ and energy intake (r = 0.388, p = 0.001).

Results of multivariable linear regression analysis are presented in Tables 3 and 4. For Model 1; body mass index, Hb level, serum albumin concentration, salt and energy intake as variables with p values below 0.10, as well as eGFR, serum sodium concentration, and protein intake as confounding factors, were included in multivariable linear regression analysis. As shown in Table 3, cerebral $rSO_2$ was independently associated with energy intake (standardized coefficient: 0.370) and serum albumin concentration (standardized coefficient: 0.236). The energy/protein index and energy/salt index were included in place of salt, protein, and energy intake as variables in Model 2 to avoid collinearity with Model 1. As a result, energy/

**Table 2. Correlation between cerebral oxygen saturation and clinical parameters, including dietary intake and nutritional parameters, in simple linear regression analysis.**

| Characteristics | Total patients n = 67 | vs. cerebral $rSO_2$ values in simple linear regression | |
|---|---|---|---|
| | | r | p value |
| Age (years) | 65.6 ± 15.6 | -0.119 | 0.338 |
| Body mass index (kg/m²) | 24.8 ± 5.2 | -0.245 | 0.045 * |
| Systolic blood pressure (mmHg) | 138 ± 18 | -0.037 | 0.764 |
| Diastolic blood pressure (mmHg) | 77 ± 14 | 0.059 | 0.633 |
| Sat $O_2$ (%) | 97.9 ± 0.7 | -0.006 | 0.961 |
| Laboratory findings | | | |
| Hb (g/dL) | 11.9 ± 1.8 | 0.271 | 0.027 * |
| eGFR (mL/min/1.73m²) | 25.5 ± 17.1 | 0.201 | 0.104 |
| Na (mEq/L) | 139 ± 3 | -0.006 | 0.963 |
| K (mEq/L) | 4.7 ± 0.6 | 0.065 | 0.602 |
| Cl (mEq/L) | 107 ± 4 | -0.130 | 0.296 |
| Total protein (g/dL) | 7.0 ± 0.6 | -0.010 | 0.938 |
| Serum albumin (g/dL) | 3.9 ± 0.4 | 0.264 | 0.031 * |
| Urinary protein excretion (g/g-Cr) | 1.0 (0.2–1.2) | | |
| ln (urinary protein excretion) | -0.8 ± 1.4 | -0.125 | 0.314 |
| Nutritional markers | | | |
| Energy intake (kcal/kg ideal BW/day) | 27.0 ± 4.2 | 0.388 | 0.001 * |
| Protein intake (g/ kg ideal BW/day) | 0.8 ± 0.2 | -0.036 | 0.775 |
| Salt intake (g/day) | 6.3 ± 2.3 | -0.228 | 0.064 |
| Energy/protein index (kcal/kg ideal BW/g-protein) | 0.7 ± 0.2 | 0.203 | 0.099 |
| Energy/salt index (kcal/kg ideal BW/g-salt) | 4.9 ± 2.1 | 0.332 | 0.006 * |

Continuous data are presented as mean ± standard deviation.

*Statistically significant.

Abbreviations: BW, body weight; eGFR, estimated glomerular filtration rate; Hb, hemoglobin; $rSO_2$, regional oxygen saturation.

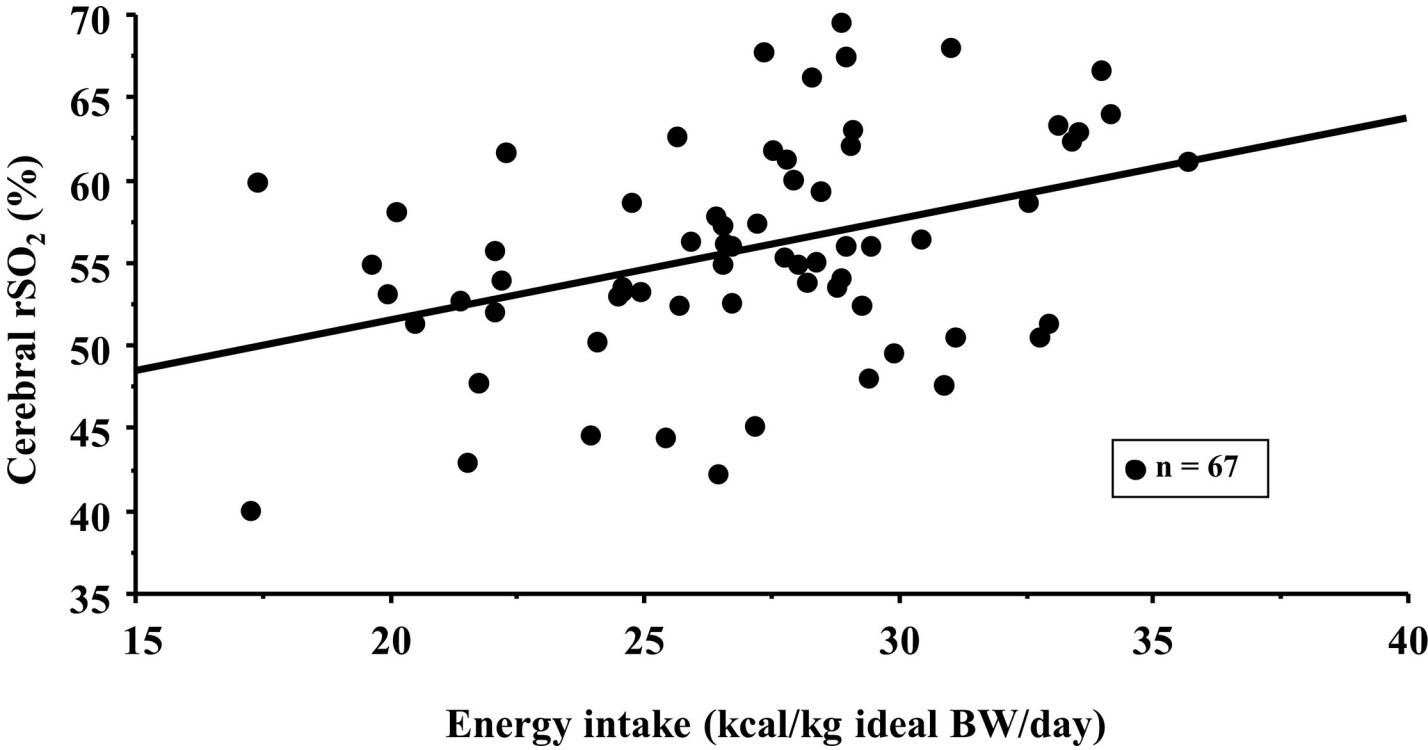

**Fig 2. Correlation between cerebral oxygen saturation and energy intake in advanced chronic kidney disease patients.** Equation of trend line (representing cerebral oxygen saturation) = 0.614 × energy intake + 39.1; r = 0.388, p = 0.001. Abbreviations: BW, body weight; $rSO_2$, regional saturation of oxygen.

salt index (standardized coefficient: 0.343) and Hb level (standardized coefficient: 0.284) were also identified as factors affecting cerebral $rSO_2$ in this study (Table 4).

## Discussion

The present study focused on the association between cerebral oxygenation and nutritional status including indices of dietary intake in CKD patients who were not receiving dialysis.

**Table 3. Multivariable linear regression analysis in Model 1 using variables including salt, protein, and energy intake as a nutritional marker: independent factors of cerebral oxygen saturation.**

| | Multivariable linear regression | |
|---|---|---|
| vs. cerebral $rSO_2$ | Standardized coefficient | p value |
| Body mass index | -0.152 | 0.201 |
| Hb | 0.205 | 0.078 |
| eGFR | 0.179 | 0.118 |
| Na | 0.052 | 0.659 |
| Serum albumin | 0.236 | 0.039 * |
| Salt intake | -0.166 | 0.155 |
| Protein intake | 0.011 | 0.923 |
| Energy intake | 0.370 | 0.002 * |

*Statistically significant.

Abbreviations: eGFR, estimated glomerular filtration rate

Hb, hemoglobin; $rSO_2$, regional oxygen saturation.

**Table 4. Multivariable linear regression analysis in Model 2 using variables including energy/protein index and energy/salt index as a nutritional marker: independent factors of cerebral oxygen saturation.**

| | Multivariable linear regression | |
|---|---|---|
| vs. cerebral rSO$_2$ | Standardized coefficient | p value |
| Body mass index | -0.144 | 0.228 |
| Hb | 0.284 | 0.014 * |
| eGFR | 0.121 | 0.417 |
| Na | 0.069 | 0.560 |
| Serum albumin | 0.191 | 0.128 |
| Energy/protein index | 0.115 | 0.409 |
| Energy/salt index | 0.343 | 0.003 * |

* Statistically significant.

Abbreviations: eGFR, estimated glomerular filtration rate

Hb, hemoglobin; rSO$_2$, regional oxygen saturation.

These results confirmed that cerebral rSO$_2$ levels are independently associated with energy intake and serum albumin concentration in Model 1 and with energy/salt index and Hb level in Model 2.

It has previously been reported that cerebral rSO$_2$ values of healthy individuals are nearly 70%, whereas those in patients undergoing HD are lower at around 50% [14,15]. Furthermore, cerebral rSO$_2$ values have been shown to decrease according to the progression of renal dysfunction [17]. In this study, cerebral rSO$_2$ values were found to lie between those of healthy individuals and patients undergoing HD, consistent with the previous report [17].

In both models for determination of modifiable factors independently associated with cerebral rSO$_2$, energy intake was found to be the most important factor. Adequate dietary intake and nutritional status have well-understood impacts on brain functions, and the mechanisms involved in the transfer of energy from foods to neurons are likely to be fundamental to the control of brain function [28]. Therefore, the effect of energy intake on cerebral oxygenation might be explained by the fact that this factor is essential for the maintenance of brain function via the energy supply to brain tissues, including cerebral microcirculation. Furthermore, it has been recently reported that the brain-gut axis is very important in the control of dietary intake [29]. Ghrelin, which is secreted primarily by epithelial cells of the stomach, stimulates food intake and is strongly associated with the regulation of energy homeostasis [30,31]. In addition, beneficial effects on vascular function and cardiovascular disease have been reported in response to ghrelin, via the stimulation of nitric oxide production and prevention of endothelial cell apoptosis [32–35]. Ghrelin might, therefore, play an important role in the maintenance of microcirculation and oxygenation in systemic tissues. The changes that occur in circulating ghrelin levels in the case of CKD and the effects of ghrelin in this context remain controversial [36,37]. However, the administration of ghrelin to patients with advanced CKD undergoing dialysis leads to increased appetite and food intake and consequent changes in energy balance [38,39]. Based on these results, ghrelin might simultaneously influence energy intake and systemic oxygenation status, including that of the brain, via the regulation of energy homeostasis and prevention of microcirculation impairment, even in patients with advanced CKD. The results presented here of the significant and positive association between cerebral rSO$_2$ and energy intake may therefore reflect the influence of the brain-gut axis, including the effects of ghrelin. However, the effects of ghrelin were not directly investigated in this study; therefore, we cannot comment on the association between cerebral oxygenation, energy intake, and the effects of ghrelin.

Salt intake has previously been reported to be associated with the progression of renal dysfunction [11,12] and cerebrovascular disease including cognitive impairments [40,41]. Recently, studies in mice have shown that high salt diets induce marked cerebral hypoperfusion and deterioration of cerebral microcirculation associated with endothelial dysregulation via the suppression of endothelial nitric oxide. This suppression was dependent on the high salt diet-induced interleukin-17 response [42], and changes in cerebral blood flow that are affected by salt intake are proposed as a new brain-gut axis. Therefore, according to the degree of increase in salt intake, cerebral oxygenation could be expected to worsen due to decreased oxygen supply induced by the deterioration of cerebral microcirculation. In this study, the mean salt intake was found to be 6.3 ± 2.3 g/day (ranging from 2.6–14.0 g/day), even after dietary education was provided, and was negatively correlated with cerebral $rSO_2$. Furthermore, a significant association between cerebral $rSO_2$ and energy/salt index was confirmed. Based on this result, salt restriction might be an approach to maintain cerebral oxygenation in addition to sufficient energy intake in the clinical setting. However, this study could not determine a significant relationship between salt intake and cerebral $rSO_2$ values; therefore, further study is needed to confirm the effect of salt intake on cerebral oxygenation and microcirculation.

Regarding the association between cerebral $rSO_2$ and nutritional parameters in this study, serum albumin concentration and Hb level were significantly associated with cerebral $rSO_2$ in multivariate linear regression analysis. Serum albumin concentration, the main determinant of colloid osmotic pressure in vessels, plays an important role in maintaining microcirculation in systemic tissues via the movement of body fluids, mainly between the vessels and interstitium [43]. Furthermore, consistent with the present study, serum albumin concentration has been reported to be significantly associated with cerebral oxygenation in patients with all stages of CKD, as well as patients undergoing HD [15,17]. In addition, Hb is an important factor in oxygen supply to the peripheral tissues and organs, including the brain; therefore, Hb level is expected to be associated with tissue $rSO_2$. Thus far, in various clinical settings including hematology [44], surgery [45], pediatrics [46–48], and HD therapy [49], cerebral $rSO_2$ has been shown to significantly increase in line with the increasing Hb levels following blood transfusion. On the other hand, it has been reported that there is no relationship between Hb concentration and cerebral $rSO_2$ values in HD patients with well-maintained Hb levels [15,17]. In this study, it is likely that Hb levels were well-maintained (the mean value was found to be 11.9 ± 1.8 g/dL); however, the values were widely distributed, from 7.1–16.0 g/dL. This study might, therefore, confirm the association between cerebral $rSO_2$ and Hb levels, because the wide distribution of cerebral $rSO_2$ values reflects the wide distribution of Hb levels.

This study had several limitations which should be noted. First, it was limited by its relatively small sample size. Second, examination of the relationship of cerebral oxygenation with cognitive function could be considered to be important. However; in this study, cognitive assessment could not be performed because of the limits of the medical examination time for each patient. Thus, we cannot comment on the association between cerebral oxygenation and cognitive function at the present time. Third, in this study, salt intake was calculated using urinary $Na^+$ excretion based on the 24-h urine collection for each patient. These values were positively correlated to those calculated in each patient's 3-day meal record (salt intake based on the 24-h urine collection: 6.3 ± 2.3 g/day vs salt intake based on each patient's 3-day meal record: 6.1 ± 1.6 g/day, r = 0.719, p< 0.001). However, due to fluctuations in daily salt intake, the values based on the 24-h urine collection may not fully reflect the constant daily salt intake for each patient. Finally, no relationships were detected between cerebral oxygenation and markers of renal function, although cerebral $rSO_2$ has been reportedly to be associated with eGFR in patients with all stages of CKD [17]. The patients included in this study mainly suffered from severe advanced CKD, and those with CKD stage 4 or 5 represented around 70% of

the cohort (47 out of 67 included patients). This proportion is significantly different to that of the previous report (40% of the study population had CKD stage 4 or 5) [17]. This might be one of the reasons for the different observations of cerebral oxygenation with regards to renal function; however, the precise reason remains unclear. Therefore, additional studies are needed to confirm the association between cerebral oxygenation and clinical parameters including dietary intake and nutritional parameters, in addition to the examination of cognitive function.

In conclusion, cerebral $rSO_2$ is affected by energy intake and the energy/salt index in addition to serum albumin concentrations and Hb levels. Therefore, sufficient energy intake with adequate salt restriction is important to prevent the deterioration of cerebral oxygenation and might contribute to the maintenance of cognitive function in addition to the prevention of renal dysfunction in CKD patients.

## Acknowledgments

We would like to thank the study participants and our hospital's staff in the Department of Nutrition.

This work was supported by a grant from the Japanese Association of Dialysis Physicians (JADP Grant 2017–9) and a grant from The Kidney Foundation, Japan (JKFB 17–4) to SO.

## Author Contributions

**Conceptualization:** Susumu Ookawara, Yoshio Kaku, Kiyonori Ito.

**Data curation:** Susumu Ookawara, Yoshio Kaku, Kiyonori Ito.

**Formal analysis:** Susumu Ookawara, Yoshio Kaku, Kiyonori Ito.

**Funding acquisition:** Susumu Ookawara.

**Investigation:** Susumu Ookawara, Yoshio Kaku, Kiyonori Ito, Kanako Kizukuri, Aiko Namikawa, Shinobu Nakahara, Yuko Horiuchi, Nagisa Inose, Mayako Miyahara, Michiko Shiina, Saori Minato, Mitsutoshi Shindo, Haruhisa Miyazawa, Keiji Hirai, Taro Hoshino.

**Methodology:** Susumu Ookawara, Yoshio Kaku.

**Project administration:** Susumu Ookawara.

**Supervision:** Susumu Ookawara, Yoshio Kaku, Kiyonori Ito, Miho Murakoshi, Kaoru Tabei, Yoshiyuki Morishita.

**Validation:** Susumu Ookawara, Yoshio Kaku, Kiyonori Ito.

**Writing – original draft:** Susumu Ookawara, Yoshio Kaku, Kiyonori Ito.

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
