## [Decision Letter · Decision Letter 0]

30 Aug 2019

[EXSCINDED]

PONE-D-19-20009

Effects of dietary intake and nutritional status on cerebral oxygenation in patients with chronic kidney disease not undergoing dialysis: A cross-sectional study

PLOS ONE

Dear Pr Ookawara,

Thank you for submitting your manuscript to PLOS ONE. After careful consideration, we feel that it has merit but does not fully meet PLOS ONE’s publication criteria as it currently stands. Therefore, we invite you to submit a revised version of the manuscript that addresses the points raised during the review process.

Besides an expert comment on your article, I have concerns as follows；

The authors discussed possible effect of ghrelin on oxygenation.  It is an acute response after meal.  In which timing after meal, did the authors evaluated brain oxygenation?

There are possible correlations with salt intake and they discussed that in animal study, salt intake affect brain oxygenation.  In animal study, salt loading is a chronic load and possibly salt induces inflammation or endothelial dysfunction.  In contrast, in the current study authors evaluated salt intake merely one day evaluation and we can not know the cohorts take salt in constant level.

Half of cohort are diabetic and two thirds of cohort is medicated for hypertension.  Is there any effect of antidiabetic drugs, control of diabetes or blood pressure?  In table 1, authors should show the distribution of antidiabetic drugs. 

We would appreciate receiving your revised manuscript by Oct 14 2019 11:59PM. To enhance the reproducibility of your results, we recommend that if applicable you deposit your laboratory protocols in protocols.io, where a protocol can be assigned its own identifier (DOI) such that it can be cited independently in the future. For instructions see: http://journals.plos.org/plosone/s/submission-guidelines#loc-laboratory-protocols

We look forward to receiving your revised manuscript.

Kind regards,

Tatsuo Shimosawa, M.D., Ph.D.

Academic Editor

PLOS ONE

Journal Requirements:

Reviewers' comments:

Reviewer's Responses to Questions

**Comments to the Author**

1. Is the manuscript technically sound, and do the data support the conclusions?

Reviewer #1: Yes

2. Has the statistical analysis been performed appropriately and rigorously? 

Reviewer #1: Yes

3. Have the authors made all data underlying the findings in their manuscript fully available?

Reviewer #1: Yes

4. Is the manuscript presented in an intelligible fashion and written in standard English?

Reviewer #1: Yes

5. Review Comments to the Author

Reviewer #1: I consider the paper to ve of a very good quality, I only have a few minor comments:

- introduction, reference 16 (Kovarova et al.): The study reported a significant rekation of rSO2 values with Montreal Cognitive Assessment, which is more specific for cognitive impairment in ESRD patients, not with MMSE

- results, lines 1 and 2: ... rSO2 values were... (there is “was” used incorrectly in the article)

- I would consider presentation of rSO2 values in different patient groups according to CKD stage interesting.

6. PLOS authors have the option to publish the peer review history of their article (what does this mean?). If published, this will include your full peer review and any attached files.

Reviewer #1: No

---

## [Author Response · Author response to Decision Letter 0]

6 Sep 2019

Response to academic editors’ and reviewers’ comments.

We appreciate your careful review and hope that we have satisfactorily addressed each of your comments in the section below.

Academic editor:

Comment 1:

The authors discussed possible effect of ghrelin on oxygenation. It is an acute response after meal. In which timing after meal, did the authors evaluate brain oxygenation?

Response 1:

Thank you for your comment. It was previously reported that ghrelin might contribute to the positive balance in energy intake and systemic circulatory stability and potentially improve systemic tissue oxygenation, including in the brain. Therefore, we discussed ghrelin’s potential effect on the energy intake and cerebral oxygenation in patients with CKD in our discussion section in the main manuscript. However, to date, few studies have investigated these relationships, and further studies will be needed to confirm ghrelin’s association with oxygenation in patients with CKD. Fasting ghrelin levels were high, whereas postprandial ghrelin levels rapidly decreased; therefore, the timing of cerebral oxygenation measurements is of great importance. As kindly suggested by the editor, we have added a statement regarding the timing of cerebral oxygenation measurements in the materials and methods section as follows:

Page 6, Lines 25-26:

“This measurement was performed approximately from 2 h to 4 h after each meal for each patient.”

Comment 2:

There are possible correlations with salt intake and they discussed that in animal study, salt intake affect brain oxygenation. In animal study, salt loading is a chronic load and possibly salt induces inflammation or endothelial dysfunction. In contrast, in the current study authors evaluated salt intake merely one day evaluation and we can not know the cohorts take salt in constant level.

Response 2:

We appreciate your thoughtful comment on this subject. As mentioned, daily salt intake fluctuates in each patient and we cannot conclude that values obtained in this study fully reflect the constant daily salt intake by each patient. However, these values were confirmed to significantly and positively correlate with those calculated in each patient’s 3-day meal record (salt intake based on the 24-h urine collection: 6.3 ± 2.3 g/day vs salt intake based on each patient’s 3-day meal record: 6.1 ± 1.6 g/day, r = 0.719, p< 0.001). To address this question, repeated evaluation of salt intake based on the 24-h urine collection would be necessary for each patient along with confirmation of these values in a clinical setting. We added a paragraph that refers to these limitations in the revised manuscript, as follows:

Page 11, Lines 22-29:

“Third, in this study, salt intake was calculated using urinary Na+ excretion based on the 24-h urine collection for each patient. These values were positively correlated to those calculated in each patient’s 3-day meal record (salt intake based on the 24-h urine collection: 6.3 ± 2.3 g/day vs salt intake based on each patient’s 3-day meal record: 6.1 ± 1.6 g/day, r = 0.719, p< 0.001). However, due to fluctuations in daily salt intake, the values based on the 24-h urine collection may not fully reflect the constant daily salt intake for each patient.”

Comment 3:

Half of cohort are diabetic and two thirds of cohort is medicated for hypertension. Is there any effect of antidiabetic drugs, control of diabetes or blood pressure? In table 1, authors should show the distribution of antidiabetic drugs. 

Response 3:

We have taken your suggestion under consideration and added the distribution of antidiabetic drugs in Table 1, as follows:

Antidiabetic medication

Insulin agent 9 (13.4)

Dipeptidyl peptidase-4 inhibitor 17 (25.4)

Insulin secretagogue 4 (6.0)

α-glucosidase inhibitor 3 (4.5)

Thiazolidinedione 3 (4.5)

Sodium-glucose cotransporter-2 inhibitor 3 (4.5)

In patients with DM, plasma glucose was 137 ± 48 mg/dL and HbA1c was 6.7 ± 0.9%. Cerebral rSO2 did not show a significant correlation with levels of plasma glucose (r = -0.091, p = 0.666) or HbA1c (r = 0.130, p = 0.565). As shown in Table 2, cerebral rSO2 did not show a significant correlation with systolic BP (r = -0.037, p = 0.764) or diastolic BP (r = 0.059, p = 0.633). Furthermore, in patients with DM, systolic BP was 138 ± 21 mmHg and diastolic BP was 75 ± 15 mmHg. For these patients, cerebral rSO2 did not show a significant correlation with systolic BP (r = 0.136, p = 0.458) or with diastolic BP (r = 0.070, p = 0.704). 

  

Reviewer 1:

Comment 1:

- introduction, reference 16 (Kovarova et al.): The study reported a significant rekation of rSO2 values with Montreal Cognitive Assessment, which is more specific for cognitive impairment in ESRD patients, not with MMSE.

Response 1:

Thank you for your careful review of our manuscript and detection of this incorrect description. We have corrected it in the Introduction section, as follows: 

Page 3, Lines 20-21:

“The results of these measurements reflect the status of cognitive impairment because of the relationship of rSO2 with the Mini-Mental State Examination scores [16] and the Montreal Cognitive Assessment test [17].”

Comment 2:

- results, lines 1 and 2: ... rSO2 values were... (there is “was” used incorrectly in the article)

Response 2:

Thank you for your diligent proofreading of our manuscript. We have now corrected the description in the results section, as follows:

Page 8, Lines 3-4:

“The mean cerebral rSO2 values of the CKD patients in this study were 55.9 ± 6.6%, and these were significantly positively correlated with Hb level, serum albumin concentration, energy intake, and energy/salt index.”

Comment 3:

- I would consider presentation of rSO2 values in different patient groups according to CKD stage interesting.

Response 3:

Thank you for your thoughtful comment. We agree with your suggestion. In this study, cerebral rSO2 values in CKD stages G1 to G3b (n = 20), G4 (n = 28), and G5 (n = 19) were 57.3 ± 6.2%, 55.9 ± 6.9%, and 53.2 ± 6.3%, respectively, and no significant differences were found among the 3 groups using a one-way ANOVA analysis (p = 0.113). However, under the assumption of an effect size of 0.25, alpha error probability of 0.05, and statistical power of 0.80, comparison among 3 groups using ANOVA would require 159 patients in total (53 patients per group). In our study, the number of the patients is currently insufficient to accurately perform these analyses. Therefore, further studies including larger patient population will be needed to confirm these results.

---

## [Decision Letter · Decision Letter 1]

25 Sep 2019

Effects of dietary intake and nutritional status on cerebral oxygenation in patients with chronic kidney disease not undergoing dialysis: A cross-sectional study

PONE-D-19-20009R1

Dear Dr. Ookawara,

We are pleased to inform you that your manuscript has been judged scientifically suitable for publication and will be formally accepted for publication once it complies with all outstanding technical requirements.

With kind regards,

Tatsuo Shimosawa, M.D., Ph.D.

Academic Editor

PLOS ONE

Additional Editor Comments (optional):

Reviewers' comments:

Reviewer's Responses to Questions

**Comments to the Author**

1. If the authors have adequately addressed your comments raised in a previous round of review and you feel that this manuscript is now acceptable for publication, you may indicate that here to bypass the “Comments to the Author” section, enter your conflict of interest statement in the “Confidential to Editor” section, and submit your "Accept" recommendation.

Reviewer #1: All comments have been addressed

2. Is the manuscript technically sound, and do the data support the conclusions?

Reviewer #1: Yes

3. Has the statistical analysis been performed appropriately and rigorously? 

Reviewer #1: Yes

4. Have the authors made all data underlying the findings in their manuscript fully available?

Reviewer #1: Yes

5. Is the manuscript presented in an intelligible fashion and written in standard English?

Reviewer #1: Yes

6. Review Comments to the Author

Reviewer #1: (No Response)

7. PLOS authors have the option to publish the peer review history of their article (what does this mean?). If published, this will include your full peer review and any attached files.

Reviewer #1: No

---

## [Editor Report · Acceptance letter]

1 Oct 2019

PONE-D-19-20009R1 

Effects of dietary intake and nutritional status on cerebral oxygenation in patients with chronic kidney disease not undergoing dialysis: A cross-sectional study 

Dear Dr. Ookawara:

I am pleased to inform you that your manuscript has been deemed suitable for publication in PLOS ONE. Congratulations! Your manuscript is now with our production department. 

With kind regards,

on behalf of

Prof. Tatsuo Shimosawa 

Academic Editor

PLOS ONE